# NanoFlow: Scalable Normalizing Flows with Sublinear Parameter Complexity

**Sang-gil Lee**      **Sungwon Kim**      **Sungroh Yoon**[*]
Data Science & AI Lab.
Seoul National University
{tkdrlf9202, ksw0306, sryoon}@snu.ac.kr

## Abstract

Normalizing flows (NFs) have become a prominent method for deep generative models that allow for an analytic probability density estimation and efficient synthesis. However, a flow-based network is considered to be inefficient in parameter complexity because of reduced expressiveness of bijective mapping, which renders the models unfeasibly expensive in terms of parameters. We present an alternative parameterization scheme called NanoFlow, which uses a single neural density estimator to model multiple transformation stages. Hence, we propose an efficient parameter decomposition method and the concept of flow indication embedding, which are key missing components that enable density estimation from a single neural network. Experiments performed on audio and image models confirm that our method provides a new parameter-efficient solution for scalable NFs with significant sublinear parameter complexity.

## 1   Introduction

Flow-based models have become a prominent approach for density estimation and generative models in recent times. These models are based on normalizing flows (NFs) [27], wherein a deep invertible model is trained with an analytically estimated likelihood of data. The model learns a bijective mapping between the data and a known prior (typically isotropic Gaussian), and its reverse operation synthesizes realistic samples from the prior. Compared with the variational autoencoder [18] and generative adversarial network [10], NFs exhibit the distinct characteristic of an exact probability density estimation from a principled maximum likelihood training. When combined with non-autoregressive coupling methods [6, 19], NFs become a powerful generative model that offers a significantly simplified training and efficient inference.

Since the introduction of the framework into neural networks, the existing studies on flow-based models have focused on improving the expressiveness of the bijective operation [2,7,12,19]. However, parameter complexity and memory efficiency are less emphasized by the research community. The small efforts to maximize the expressiveness *under a specified amount of capacity of the neural network* has recently become problematic when expanding a flow-based model for real-world applications. A notable example is the waveform synthesis model [15, 26]. Although the aforementioned studies have achieved audio generation faster than real-time (thereby removing the slow inference bottleneck of WaveNet [29]), they resulted in an increase in the number of parameters by an order of magnitude, which is unfeasibly expensive in terms of memory. Hence, building a complex, scalable, and *memory-efficient* flow-based model remains challenging.

This scenario raised a question: *Is it true that NFs require a significantly larger network capacity to perform expressive bijections, or is the representational power of deep neural networks inefficiently*

---

[*]Corresponding author

*utilized?* We argue that studies regarding NFs should consider the parameter complexity, where the expressiveness of multiple flows is not necessarily accompanied by a linearly growing number of parameters.

In this study, we challenge the typical assumption in building flow-based models and aim to decouple the required number of parameters and the expressiveness of multiple bijective operations for flow-based models. We present NanoFlow, an alternative parameterization scheme for NFs that operates on a single neural density estimator. Because the shared density estimator is applied to multiple stacks of flows, the parameter requirement is no longer proportional to the number of flows, and the memory footprint is significantly reduced. Consequently, NanoFlow can consistently improve its expressiveness by stacking flows without sacrificing parameter efficiency.

Our results indicated that using a conventional notion of weight sharing did not yield a good performance on flow-based models, which nullifies the potential benefits. To achieve the concept of a shared neural density estimator, we demonstrate several parameter-efficient solutions for increasing the flexibility of NanoFlow. We show that decomposing a deep hidden representation estimated by the shared neural network and the projected densities from the representation can significantly enhance the expressiveness of NanoFlow with the addition of a few parameters. Furthermore, we also demonstrate that conditioning the shared estimator with our flow indication embedding can remedy the modeling difficulties of multiple densities from a single estimator without dissatisfying any invertibility constraints.

Additionally, we provide a deeper analysis of the condition under which our method yields the a higher number of benefits. Specifically, we assess the effectiveness of the single density estimator by varying the amount of autoregressive structural bias into the model. Our results demonstrate that our method performs best on bipartite flows, which provides an expanded narrative on our belief regarding the performance gap between non-autoregressive and autoregressive models. In summary, our study is the first to focus on a systematic assessment for enabling scalable NFs with an almost constant parameter complexity.

## 2 Background

NFs learn the bijective mapping between data and a known prior. The prior is typically constructed as an isotropic Gaussian, and the reverse of the bijective mapping can synthesize the data from the noise sampled from the prior. Formally, NFs learn the bijective function $f(\boldsymbol{x}) = \boldsymbol{z}$, which transforms a complex data probability distribution $P_{\boldsymbol{X}}$ into a simple known prior $P_{\boldsymbol{Z}}$ with the same dimension. We can analytically compute the probability density of real data $\boldsymbol{x}$ using the change of variables formula:

$$\log P_{\boldsymbol{X}}(\boldsymbol{x}) = \log P_{\boldsymbol{Z}}(\boldsymbol{z}) + \log|\det(\frac{\partial f(\boldsymbol{x})}{\partial \boldsymbol{x}})|, \tag{1}$$

where $\det(\frac{\partial f(\boldsymbol{x})}{\partial \boldsymbol{x}})$ is a Jacobian determinant of the function $f(\boldsymbol{x}) = \boldsymbol{z}$. To enhance the expressiveness of $f$, NF models decompose the function into multiple flows as follows:

$$f = f^K \circ f^{K-1} \circ ... \circ f^1(\boldsymbol{x}), \tag{2}$$

where $K$ is the number of flows defined by the model. Using the notations $\boldsymbol{x} = \boldsymbol{z}^0$ and $\boldsymbol{z} = \boldsymbol{z}^K$, each $f^k(\boldsymbol{z}^{k-1}) = \boldsymbol{z}^k$ learns the intermediate densities between $\boldsymbol{x}$ and $\boldsymbol{z}$, and $\log P_{\boldsymbol{X}}$ can be re-expressed as follows:

$$\log P_{\boldsymbol{X}}(\boldsymbol{x}) = \log P_{\boldsymbol{Z}}(\boldsymbol{z}) + \sum_{k=1}^{K} \log|\det(\frac{\partial f^k(\boldsymbol{z}^{k-1})}{\partial \boldsymbol{z}^{k-1}})|. \tag{3}$$

Because the determinant typically requires $O(n^3)$ computing time (where $n$ is the dimension of the data), NF models are designed to maintain a triangular Jacobian [5,17,23]. By maintaining a triangular Jacobian, the determinant becomes easy to compute, and the model becomes computationally tractable for both forward and inverse functions.

Our mathematical notation for the coupling transformation follows that of the WaveFlow [25]. Although the study focused on waveform synthesis, it provides a unified view from bipartite to autoregressive flows, which subsumes a wide range of flow-based models. We note that [23] also provides a relevant analysis regarding the relationship between autoregressive and bipartite flows.

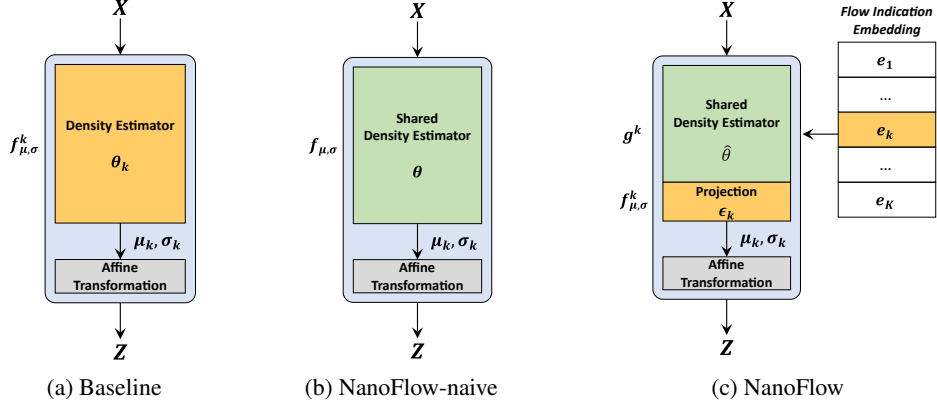

(a) Baseline          (b) NanoFlow-naive          (c) NanoFlow

Figure 1: High-level overview of NanoFlow. (a) Conventional NFs employ separate neural networks as a density estimator for each flow. (b) NanoFlow-naive shares an entire part of the neural network for density estimation with multiple flow steps. (c) NanoFlow decomposes the estimator into two parts—one for the deep shared latent space representation augmented by flow indication embedding, and another for separate projection layers employed to each flow.

Formally, for training data $\boldsymbol{x}$, assume that we split $\boldsymbol{x}$ into $G$ groups as $\{\boldsymbol{X}_1, ..., \boldsymbol{X}_G\}$. The model is trained to learn the bijective mapping between $\boldsymbol{X}$ and a prior $\boldsymbol{Z}$ with the same dimension. This is achieved by applying an affine transformation $f : \boldsymbol{X} \rightarrow \boldsymbol{Z}$ which models a sequential dependency between the grouped data as follows:

$$\boldsymbol{Z}_i = \sigma_i(\boldsymbol{X}_{<i}; \theta) \cdot \boldsymbol{X}_i + \mu_i(\boldsymbol{X}_{<i}; \theta), \quad i = 1, ..., G, \tag{4}$$

where $\boldsymbol{X}_{<i}$ refers to all the partitions of the data before the $i$-th group, $\boldsymbol{X}_i$; $\sigma$ and $\mu$ are the scale and shift variables estimated by the neural networks, respectively. From the sampled noise $\boldsymbol{Z}$, the inverse of the affine transformation $f^{-1} : \boldsymbol{Z} \rightarrow \boldsymbol{X}$ sequentially generates $\boldsymbol{X}$ as follows:

$$\boldsymbol{X}_i = \frac{\boldsymbol{Z}_i - \mu_i(\boldsymbol{X}_{<i}; \theta)}{\sigma_i(\boldsymbol{X}_{<i}; \theta)}, \quad i = 1, 2, ..., G. \tag{5}$$

The model becomes a purely autoregressive flow when $G = dim(\boldsymbol{x})$. Conversely, the equations theoretically represent bipartite flows when $G = 2$. Increasing the number of groups introduces a higher amount of autoregressive structural bias into the model, at a cost of $O(G)$ inference latency.

As previously mentioned, the entire bijective function $f : \boldsymbol{X} \rightarrow \boldsymbol{Z}$ is decomposed into $K$ flows as $f = f^K \circ f^{K-1} \circ ... \circ f^1(\boldsymbol{X})$, where we use the notation $\boldsymbol{X} = \boldsymbol{Z}^0$ and $\boldsymbol{Z} = \boldsymbol{Z}^K$. Each $f^k : \boldsymbol{Z}^{k-1} \rightarrow \boldsymbol{Z}^k$ is parameterized by separate neural networks $\theta^k$, whereas each $\theta^k$ estimates the intermediate density of $\boldsymbol{Z}^k$ by computing $\sigma^k$ and $\mu^k$ for the flow operation. For clarity, we consider the notation of $\boldsymbol{X}$ as the input and $\boldsymbol{Z}$ as the output for $f^k$. We re-write $f^k$ for completeness as follows:

$$\boldsymbol{Z}_i = \sigma_i(\boldsymbol{X}_{<i}; \theta^k) \cdot \boldsymbol{X}_i + \mu_i(\boldsymbol{X}_{<i}; \theta^k). \tag{6}$$

The above formula describes the affine transformation as a bijective coupling operation. Various other classes of coupling exist in the literature [7, 12].

## 3 NanoFlow

In this section, we present NanoFlow, a new alternative parameterization scheme for a flow-based model (Figure 1). The main goal of NanoFlow is to decouple the expressiveness of the bijections and the parameter efficiency of density estimation from neural networks. We initially describe a core change in the design of the neural architecture by decomposing the parameters for neural density estimation and sharing parameters across flows.

### 3.1 Parameter sharing and decomposition

We reformulated $f_{\mu,\sigma}^k$ using a single shared neural network $f_{\mu,\sigma}$, parameterized by $\theta$. Based on this framework, all $\sigma^k$ and $\mu^k$ for each flow were estimated by the shared $f_{\mu,\sigma}$. This formulation

Table 1: Comparison of parameterization scheme of $f^k$ between methods for bijection. $K$ is the total number of flows defined by the model, and $|\bullet|$ is the number of parameters of the neural network with the designated letters.

| Method | $f^k : \boldsymbol{X}_i \to \boldsymbol{Z}_i = \sigma_i^k \cdot \boldsymbol{X}_i + \mu_i^k, i = 1, ..., G$ | Parameters |
|---|---|---|
| WaveFlow (baseline) | $\mu_i^k, \sigma_i^k = f_{\mu,\sigma}^k(\boldsymbol{X}_{<i}; \theta^k)$ | $\sum_{k=1}^K |\theta^k|$ |
| NanoFlow-naive | $\mu_i^k, \sigma_i^k = f_{\mu,\sigma}(\boldsymbol{X}_{<i}; \theta)$ | $|\theta|$ |
| NanoFlow-decomp | $\mu_i^k, \sigma_i^k = f_{\mu,\sigma}^k(g(\boldsymbol{X}_{<i}; \hat{\theta}); \epsilon^k)$ | $|\hat{\theta}| + \sum_{k=1}^K |\epsilon^k|$ |
| NanoFlow (proposed) | $\mu_i^k, \sigma_i^k = f_{\mu,\sigma}^k(g^k(\boldsymbol{X}_{<i}; \hat{\theta}, \boldsymbol{e}^k); \epsilon^k)$ | $|\hat{\theta}| + \sum_{k=1}^K (|\epsilon^k| + |\boldsymbol{e}^k|)$ |

can reduce the number of parameters by a fraction of the number of flows by $\frac{1}{K}$, and we call this variant, the NanoFlow-naive. However, as our experimental results suggest, this aggressive re-use of parameters is unsuitable for modeling multiple densities that suffer from severe degradation in performance, as it completely nullifies the potential benefit of the $O(1)$ memory footprint.

Based on these observations, we propose to relax the constraint of the shared estimator by decomposing the shared model into a network that computes a hidden representation and a projection layer that estimates the densities. The function is decomposed into $f_{\mu,\sigma}^k \circ g$, where $g(\cdot; \hat{\theta})$ is the shared estimator parameterized by $\hat{\theta}$, excluding the projection layer, that is, each $f_{\mu,\sigma}^k$ has separate parameters for the projected intermediate density by computing $\sigma^k$ and $\mu^k$.

Assuming that $g$ has sufficient capacity for density estimation, the projection layer can be as shallow as a $1 \times 1$ convolution; hence, the number of parameters is negligible in comparison with $\hat{\theta}$. Using this decomposition, we can construct NanoFlow with an arbitrary number of flows. Interestingly, this alternative scheme was pivotal for achieving the competitive performance of NanoFlow. We observed that the likelihood of the data continuously increased as we stacked additional flows without sacrificing the efficiency of weight sharing. Hence, we can re-write $f^k$ as follows:

$$\boldsymbol{Z}_i = \sigma_i(\boldsymbol{X}_{<i}; \hat{\theta}, \epsilon^k) \cdot \boldsymbol{X}_i + \mu_i(\boldsymbol{X}_{<i}; \hat{\theta}, \epsilon^k), \tag{7}$$

where $\epsilon^k$ is the parameter of the separate projection layer assigned to each flow.

## 3.2 Flow indication embedding

Even when the parameter decomposition described above is used, the shared estimator $g$ must learn multiple intermediate densities of bijective transformations without context. Hence, we introduce a key missing module, which we name flow indication embedding, to enable the shared model to simultaneously learn multiple bijective transformations. Because the flow-based model is based on the bijective function, the embedding must be available *a priori* for application to the reverse operation.

For each $f^k$, we define an embedding vector $\boldsymbol{e}^k \in \mathbb{R}^D$, where $D$ is the dimension of the embedding. Subsequently, we feed the embedding to the shared model $g(\cdot; \hat{\theta})$ as an additional context. From the embedding $\boldsymbol{e}^k$, we can further guide $g(\cdot; \hat{\theta})$ to learn multiple intermediate densities with minimal addition of parameters, transforming it into $g^k(\cdot; \hat{\theta}, \boldsymbol{e}^k)$. Because the order of flow operations is pre-defined, we can use $\boldsymbol{e}^k$ by feeding it to the shared estimator in the reverse order during inference, that is, our embedding does not dissatisfy any invertibility constraints.

The optimal injection of the embedding into $g^k$ may depend on the neural architecture. We investigated three candidates: 1. concatenative embedding, in which we augment the input with the embedding vector at the start of each flow; 2. additive bias, in which for each layer inside $g^k$, $\boldsymbol{e}^k$ provides a channel-wise bias projected from additionally defined $1 \times 1$ convolutional layers; 3. multiplicative gating, in which we employ independent per-channel scalars inside $g^k$ for each flow that controls the propagation of the convolutional feature map according to the specified flow steps. Note that the aforementioned methods involve a negligible number of additional parameters, do not dissatisfy the invertibility, and impose a minimal effect on the inference latency. See Appendix A for a more detailed description.

Table 2: Model comparison. We report the number of model parameters in millions (M), a log-likelihood (LL) on the test set, a subjective five-scale mean opinion score (MOS) on naturalness with 95 % confidence interval, and a synthesis speed using a single Nvidia V100 GPU with half-precision arithmetic. MOS on ground-truth audio is $4.58 \pm 0.06$.

| Method | Channels | Parameters (M) | LL | MOS | kHz |
|---|---|---|---|---|---|
| WaveFlow (Re-impl) | 128 | 22.336 | 5.2059 | $4.11 \pm 0.08$ | 347 |
| WaveFlow (Re-impl) | 64 | 5.925 | 5.1357 | $3.52 \pm 0.09$ | 828 |
| NanoFlow-naive | 128 | 2.792 | 5.1247 | $3.23 \pm 0.09$ | 376 |
| NanoFlow | 128 | 2.819 | 5.1586 | $3.63 \pm 0.09$ | 362 |
| NanoFlow (K=16) | 128 | 2.845 | 5.1873 | $3.82 \pm 0.08$ | 186 |

Table 1 summarizes the parameterization scheme of $f^k$ and its complexity. The parameter efficiency of NanoFlow is due to employing a single neural density estimator, $g$, for multiple flow operations. NanoFlow-naive incorporates a conventional weight sharing and NanoFlow-decomp relaxes the constraint of intermediate density estimations by employing separate $\epsilon^k$ for each flow. The final proposed model, NanoFlow, further increases the parameter efficiency of $g^k$ by incorporating the flow indication embedding $e^k$. We emphasize that $\hat{\theta}$ embodies the majority of the parameters from the model.

## 4 Experiments

In this section, we present a systematic assessment of the effectiveness of NanoFlow. We initially present the experimental results from an audio generative model with WaveFlow [25] as the baseline architecture, combined with an extensive ablation study. Next, we provide a likelihood ratio analysis of NanoFlow by varying the amount of autoregressive structural bias into both models, which evaluates the conditions under which NanoFlow yields more benefits. Finally, we investigate the generalizability of our methods by performing density modeling on the image domain, with Glow [19] as the reference model.

### 4.1 Audio generation results

For the performance evaluation of waveform generation, we used the LJ speech dataset [14], which is a 24-h single-speaker speech dataset containing 13,100 audio clips. We used the first 10% of the audio clips as the test set and the remaining 90% as the training set. We used the audio preprocessing and mel-spectrogram construction pipeline provided by the official WaveGlow implementation [26]. Specifically, we used an 80-band log-scale mel-spectrogram condition with an FFT size of 1,024, a hop size of 256, and a window size of 1,024. We used a maximum frequency of 8,000Hz for the STFT without audio volume normalization, and we set the noise distribution to $\boldsymbol{Z}_i \sim \mathcal{N}(0, 1)$, which is the default setting from the open-source WaveGlow.

We used the default architecture described in [25] with $G = 16$ for WaveFlow and NanoFlow. We constructed the models with eight flows unless otherwise specified and used the permutation strategy of reversing the order of the group dimensions per flow for both models. Our selection of flow indication embedding is a combination of additive bias and multiplicative gating, as WaveNet-based [29] architecture features a natural method of utilizing additive bias as global conditioning augmented by a gated residual path [30]. We used $D = 512$ for $e^k \in \mathbb{R}^D$ in the default eight-flows model and $D = 1024$ in the 16-flows variant.

We trained all models for 1.2 M iterations with a batch size of eight and an audio clip size of 16,000, using an Nvidia V100 GPU. We used the Adam optimizer [16] with an initial learning rate of $10^{-3}$, and we annealed the learning rate by half for every 200 K iterations. For the evaluation, we applied checkpoint averaging over 10 checkpoints with 5 K iteration intervals. We sampled the audio at a temperature of 1.0.

Table 2 shows an objective performance measure of log-likelihood (LL) on the test set as well as a subjective and relative audio quality evaluation with a five-scale mean opinion score (MOS) on

Table 3: LL ratio results with varying amount of autoregressive structural bias on the number of groups. Lower values indicate higher similarity in probability density modeling performance between the two models.

| Number of Groups (G) | 4 | 8 | 16 | 32 | 64 |
|---|---|---|---|---|---|
| LL    WaveFlow (5.96 M) | 4.9785 | 5.0564 | 5.1241 | 5.141 | 5.1586 |
| NanoFlow (0.75 M) | 4.9513 | 5.0271 | 5.0797 | 5.0927 | 5.111 |
| LL ratio | **0.0272** | 0.0293 | 0.0444 | 0.0483 | 0.0476 |

naturalness using the Amazon Mechanical Turk. Furthermore, we provide the audio synthesis speed in kilohertz using a single Nvidia V100 GPU with half-precision arithmetic.

The results show that our method can synthesize waveforms with a slight quality degradation against the baseline while only using approximately $1/8$ of the parameters. However, the NanoFlow-naive failed to perform competitively even against a 64-channel variant of WaveFlow. This suggests that for flow-based models, a strict constraint of $O(1)$ memory requirement severely degenerates the modeling capability. NanoFlow-decomp performed slightly better than NanoFlow-naive with a likelihood score of 5.13, which was still insufficient as a competitive alternative.

On the contrary, NanoFlow provided significantly enhanced expressiveness, with a negligible number of additional parameters from the decomposition technique with flow indication embedding. By stacking double the steps of flows, we further verified that the enhanced expressiveness of the flows was no longer proportional to the capacity of the deep generative model. Consistent with the results from a previous work [25], we observed that the subjective MOS scores exhibited good alignment with the objective likelihood scores.

## 4.2 Likelihood ratio analysis with autoregressive structural bias

Our reference model, WaveFlow [25], provided a unified view of the expressiveness of flow-based models by incorporating a fixed amount of autoregressive structural bias into the architecture. The model provides a hybrid method in which the autoregressive bias is proportional to the number of group dimensions. In this section, we provide an expanded narrative on the performance gap between the non-autoregressive and autoregressive flows by adjusting the amount of bias for both WaveFlow and NanoFlow. We trained each model with 64 channels for 500 K iterations with a batch size of two for varying degrees of the group dimension. We used $D = 128$ for the NanoFlow embedding.

Table 3 quantitatively shows the expressiveness of autoregressive bias. As we enforce a higher amount of the autoregressive structure into the model, we can achieve a more expressive model under the same network capacity. However, this is at the expense of sequential inference, which has been reported in previous studies [19, 22, 29].

In addition, we provide the LL ratio between WaveFlow and NanoFlow, where we measure the gap in modeling capability by introducing a shared neural density estimator. Most importantly, we observed a nearly monotonic decrease in the performance gap of NanoFlow as we decreased the number of groups. This further provides an insight into our effectiveness in utilizing the capacity of the deep generative network. If we impose a lower amount of the explicit dependency between partitions of data, we can extract a deep shared latent representation that is easier to manipulate by our flow indication embedding. In other words, we can expect a wide range of flow-based models with bipartite coupling to benefit significantly from the parameterization scheme of NanoFlow.

## 4.3 Image density modeling results

To demonstrate that our method is applicable to any configuration of NF and data domains, we assessed the effectiveness of NanoFlow's parameterization scheme to Glow [19]. We used the training configurations of an open-source implementation as described in [19]. We trained Glow, NanoFlow, and its ablations on the CIFAR10 dataset for 3,000 epochs, where all model configurations reached saturation in performance. We used 256 channels and a batch size of 64 for all configurations for an extensive ablation study under a fixed computational budget.

Table 4: Unconditional image density estimation results with bits per dimension (bpd) on CIFAR10 under uniform dequantization. Results with † were taken from the existing literature [9].

| Method | Parameters (M) | bpd |
|---|---|---|
| Glow (256 channels) | 15.973 | 3.40 |
| Glow (512 channels) [19] † | 44.235 | 3.35 |
| Glow-large | 287.489 | 3.30 |
| RQ-NSF (C) [7] † | 11.8 | 3.38 |
| FFJORD [11] † | 1.359 | 3.40 |
| MintNet [28] † | 27.461 | 3.32 |
| Flow++ [12] † | 32.3 | 3.28 |
| ResidualFlow [2] † | 25.174 | 3.28 |
| NanoFlow-naive | 9.263 | 3.40 |
| NanoFlow-decomp | 9.935 | 3.32 |
| NanoFlow | 10.113 | **3.27** |
| NanoFlow (K=48) | 10.718 | **3.25** |

Because NanoFlow is designed to leverage the shared density estimator with sufficient capacity, we increased the number of convolutional layers to six, and modified the kernel size to $3 \times 3$ for all layers. We changed the kernel size of the separate density projection layers to $1 \times 1$ to maintain the nearly constant memory footprint of NanoFlow. We refer to the model with this modified architecture without the application of our method as Glow-large. This model serves as an upper bound on modeling performance, but the parameter complexity is increased. We trained the original model with the exact network topology from [19] together with Glow-large to completely assess the capability of NanoFlow. Because Glow uses a multi-scale architecture [6], NanoFlow is applied by sharing the estimator separately for each scale. We used concatenative embedding together with multiplicative gating as the flow indication embedding. For $e^k \in \mathbb{R}^D$, we used $D = 64$ for the default 32 flows per scale, and $D = 192$ for a scaled-up model with 48 flows per scale.

As presented in Table 4, we observed that the reference Glow model scaled with a higher network capacity, at the cost of the increased parameters and decreased return. NanoFlow-naive failed to perform competitively, even with the increased capacity of the shared estimator. This suggests that even if a more powerful neural network is introduced, a critical bottleneck exists when modeling multiple flows from the single model without applying our method.

Unlike the waveform synthesis results, applying only the decomposition technique was sufficient to outperform NanoFlow-naive by a large margin. The performance was further improved using flow indication embedding. NanoFlow with the default number of flows (32 steps per scale) exhibited better performance than Glow-large, which has more than 28 times more parameters. This illustrates that in NFs, leveraging the shared neural network would be easier to train and more scalable with better inductive bias, provided with proper methods as shown by NanoFlow, than employing separate estimators where each neural network should learn the intermediate probability densities from scratch.

When we scaled up the model to 48 flows per scale, we observed an additional gain in performance from the shared estimator, further confirming the scalability of the proposed method. NanoFlow was able to achieve competitive performance compared to studies with more complex non-affine coupling [2, 7, 12, 28], indicating potential benefits of deep and shared latent representation. Overall, the density estimation results with bits per dimension were consistent with the audio generation results. The effectiveness of our method was further highlighted in this setup with bipartite coupling, which further confirms our findings from the likelihood ratio analysis in the preceding section. See Appendix B for the additional results and Appendix C for the sampled images from the models.

## 5 Related Work

### 5.1 Improving coupling transformations

Since the introduction of NFs into neural networks [5, 27], most studies have focused on composing a flexible bijection for better expressiveness [6, 7, 12, 13, 19, 21, 23]. Building a more complex bijection

can also achieve better memory efficiency by attaining the desired level of complexity under fewer flow operations. Our study provides an orthogonal perspective on this topic with a specific focus on the parameterization of a scalable NF *under a specified network capacity*, where we systematically assess the feasibility of employing a single shared neural density estimator for multiple flow steps. Because our parameterization scheme is agnostic to any setup of flow-based models and coupling operator, we can apply any off-shelf bijections into our framework, together with improved methods for training NFs [12].

## 5.2 Parameter sharing

The concept of parameter sharing has been previously studied in various domains, from the core foundation of the design principle of convolutional and recurrent neural networks to parallel sequence models, such as the Transformer [3, 20]. The most notable example is [20] in the natural language processing domain, which demonstrated a significantly reduced memory footprint of BERT [4] using a cross-layer parameter sharing of the self-attention block. We investigated the effectiveness of the weight-sharing concept on different granularities for flow-based models. We applied parameter sharing on a *model level*, where a shared neural density estimator was applied to multiple stages of bijective transformation that performed bijective operations. Contrary to [20], our study revealed the following findings: in NFs, sharing an entire block failed to competitively model the probability density, whereas minimal relaxation from the decomposition was critical to the performance.

It is noteworthy that continuous-time normalizing flows (CNF) [1, 11] features a form of the "shared" neural network $f$. The central difference between CNF and NanoFlow (and non-continuous NFs in general) is that CNF formulates the transformation by an ordinary differential equation (ODE) with an iterative evaluation of $f$ to reach an error tolerance of the adaptive ODE solver, whereas NFs directly model pre-defined steps of transformation with $f_k$ (or $f$ in NanoFlow) with a single network pass. The effectiveness and potential benefits of the shared $f$ outside the ODE-based CNFs are yet to be studied in the literature, which we aim to systematically address with NanoFlow.

## 6 Discussion

In this study, we presented an extensive and systematic analysis of the feasibility and potential benefits of using a single shared neural density estimator for multiple flow operations. Based on the analysis, we developed a novel parameterization scheme called NanoFlow, which enabled scalable NFs with a nearly constant memory complexity and competitive performance as both a generative and a density estimation model, owing to the compact network capacity. This enables direct control over the tradeoff between expressiveness and inference latency, which is beneficial in domains where compact parameterization is desired. The target performance can be explicitly designed using NanoFlow as a building block depending on the task requirements, which can be useful for practitioners who incorporate NFs into applications.

The decomposed view on building flow-based models with NanoFlow suggests that two directions can be endeavored in future research: composing more expressive bijections, which has been the primary focus in existing literature, and building an optimized neural density estimator that can potentially provide a more adaptive computation path leveraged by flow indication embedding. Furthermore, these proposed future studies can be expanded from [12], which investigated better neural architecture designs for building flow-based models using self-attention for the estimator. Combined with increasing evidence in other research domains applying similar architecture [20], we expect the self-attention-based estimator to provide more expressive density estimations [8, 24], where the attention mechanism could be directly augmented from flow indication embedding. We leave this research direction for future works.

In summary, NanoFlow, which is a bijection-agnostic and generalized solution that achieves significant savings in network capacity, provides an alternative method for parameterizing NFs. Extensive experiments on real-world data domains have provided deep insights into the relationship between the capacity of deep generative models and the expressiveness of flow operations, along with possible future research directions. We hope that the modular scheme of NanoFlow will motivate researchers to further develop flexible and scalable flow-based models.

## Broader Impact

The main motivation of this study was to observe a major hurdle in incorporating a powerful generative capability of NFs to various application domains, where we need significantly larger neural network capacity to reach the desired level of performance. As our work would impact the practicality of NFs as a mainstream probabilistic toolkit, practitioners should be cautious about possible misrepresentations of our flow indication embedding methods depending on how one further augments the embedding to specific tasks of interest.

In particular, although we demonstrated that our flow indication embedding is domain agnostic and independent variables, it is possible to incorporate task-specific priors into our framework, which can potentially achieve better control of the latent space. By contrast, there is a risk of potential misinterpretation of the embedding, together with the latent space, from biases inside the dataset. Because NFs have exact latent spaces that can be useful for downstream tasks such as facial manipulation [19], it would have a higher chance of direct exposure to various levels of biases. This could result in a potential exploitation of our embedding methods as an explainable or predictive embedding vector of the biased aspects that could be inherent in the data. Considering these possible directions for the downstream applications of NFs, one should be cautious about extrapolating our embedding scheme in attempts to build improved embedding methods for the target tasks, particularly when leveraging priors into the independent variables we demonstrated.

## Acknowledgements

We thank Wei Ping for helpful discussion and feedback on implementation details of WaveFlow [25] model. We also thank Heeseung Kim for careful proofreading. This work was supported by the BK21 FOUR program of the Education and Research Program for Future ICT Pioneers, Seoul National University in 2020 and the National Research Foundation of Korea (NRF) grant funded by the Korea government (Ministry of Science and ICT) [No. 2018R1A2B3001628].

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
