[Supplementary Material]

# Appendix

## A  Implementation details of flow indication embedding

In this section, we describe the implementation details of flow indication embedding used in this study, depending on the architecture. We used additive bias and multiplicative gating for WaveFlow-based experiments and concatenative embedding and multiplicative gating for Glow-based experiments.

**Concatenative embedding.** At the start of each flow, we concatenated the input $\boldsymbol{X}$ with $\boldsymbol{e}^k$ as the augmented representation as follows:

$$\boldsymbol{X}_{cat} = Concatenate(\boldsymbol{X}, \boldsymbol{e}^k). \tag{8}$$

The $\boldsymbol{e}^k$ was reshaped to match the shape of the input for each flow. For Glow-based experiments, we reshaped $\boldsymbol{e}^k$ as $\boldsymbol{e}^k \in \mathbb{R}^{\hat{C} \times height \times width}$, where $\hat{C} = \frac{D}{height \times width}$, and performed concatenation along the channel-axis.

**Additive bias.** We used the notation $h^{k,l} \in \mathbb{R}^{H,\cdot}$ as the hidden representation of the $l$-th layer from the $k$-th flow, where $H$ is the number of hidden channels of the neural network. We applied the channel-wise additive bias to $h^{k,l}$ using a single fully connected layer for projection as follows:

$$\tilde{h}^{k,l} = h^{k,l} + W^l \boldsymbol{e}^k, \tag{9}$$

where $W^l \boldsymbol{e}^k \in \mathbb{R}^H$. After training, we can cache the projected bias from the embedding as the final network parameters and discard the projection weights. The reported parameter count in Table 2 is obtained from the trained model with the projection weights discarded.

**Multiplicative gating.** We performed multiplicative gating to $h^{k,l}$ by employing a vector $\delta^{k,l} \in \mathbb{R}^H$ as follows:

$$\hat{h}^{k,l} = \exp(\delta^{k,l}) \cdot h^{k,l}. \tag{10}$$

$\delta^{k,l}$ was initialized to zero to initially perform the identity. For WaveFlow-based experiments, we applied additive bias followed by multiplicative gating. For Glow-based experiments, we applied multiplicative gating before applying the ReLU activation.

## B  Additional experimental results

In this section, we provide additional experimental results.

**Effect of the number of shared layers.** We measured a tolerance to the decreased network capacity of NanoFlow by varying the number of the shared layers of the network. We used the 8-layer WaveFlow with 64 residual channels as the non-shared baseline, and trained NanoFlow-naive and NanoFlow by partially replacing the layers from the bottom (i.e. closer to the input) with the shared weights. We trained all models with the batch size of two for 600 K iterations under the same learning rate schedule as described in Section 4.1. We used $D = 128$ for $\boldsymbol{e}^k$ of NanoFlow.

Figure 2 shows differences in the performance drop from the varied amount of the decreased network capacity. We can see that NanoFlow-naive degraded its performance more significantly than our final model, which indicates that the proposed technique provided better tolerance to the decreased capacity of the network.

**Compatibility beyond the affine coupling.** Our main experiments used WaveFlow [25] and Glow [19], where both models used affine coupling for the transformation. We show that NanoFlow is not restricted to a specific choice of the bijection by replacing the affine coupling for WaveFlow-based models with rational-quadratic splines (RQ-NSF) [7]. We trained both WaveFlow and NanoFlow with the same training strategy as described in Section 4.1. We used the the following hyperparameters for the rational-quadratic spline: the number of bins of 32 and the tail bound of 5. We experienced unpleasing popping sounds from the generated audio for all models if we set these values lower.

The likelihood results from Table 5 show that NanoFlow + RQ-NSF performed slightly better than the default affine coupling, whereas the high-capacity WaveFlow scored worse likelihood. We are not drawing any conclusive claim regarding the different classes of the bijection based on these

Figure 2: Analysis on the effect of the number of shared layers.

Table 5: Additional results of using non-affine coupling with rational-quadratic splines.

| Method | Params (M) | LL |
|---|---|---|
| WaveFlow | 22.336 | 5.2059 |
| NanoFlow | 2.819 | 5.1586 |
| WaveFlow + RQ-NSF | 22.432 | 5.1866 |
| NanoFlow + RQ-NSF | 2.915 | 5.1614 |

Table 6: Results when applying the method to the reference network topology of Glow model (NanoFlowAlt), evaluated at 600 epochs.

| Method (600 epochs) | Params (M) | bpd |
|---|---|---|
| Glow (256 channels) | 15.973 | **3.44** |
| NanoFlowAlt-naive | 0.778 | 3.75 |
| NanoFlowAlt-decomp | 6.783 | 3.54 |
| NanoFlowAlt | 6.961 | 3.53 |
| NanoFlowAlt (K=48) | 10.319 | 3.51 |

observations as we have not performed an exhaustive hyperparameter search and training schedule. However, the results indicate that NanoFlow is not restricted to the particular coupling and can be applied to various other classes of flows.

**Caveats**. As demonstrated in the main experimental results, NanoFlow is designed for leveraging the rich representational power of the deep neural network. In other words, a careful allocation of the parameters is required under NanoFlow framework, where the shared estimator should have sufficient capacity, while keeping the non-shared projection layers lightweight.

We additionally show a negative result when the aforementioned caveats are not met. We trained the NanoFlow variants of Glow with the exact same network topology: from $3 \times 3$ conv $\rightarrow 1 \times 1$ conv $\rightarrow 3 \times 3$ projection conv layers per flow, NanoFlowAlt shared the first two layers and used the separate $3 \times 3$ projection conv. Results showed that the model performed significantly worse than the baseline architecture, even though NanoFlowAlt (K=48) has similar network size (10 M) to our main result. This indicates that we have to assure that the shared neural density estimator possesses the sufficient capacity.

# C Samples generated from image models

(a) Glow (bpd = 3.40)

(b) Glow-large (bpd = 3.30)

(c) NanoFlow-naive (bpd = 3.40)

(d) NanoFlow (K = 48) (bpd = 3.25)

Figure 3: Unconditional samples generated from image models in Section 4.3 trained on CIFAR10. The temperature was set to 1.0. Models with lower bpd tended to generate sharper and detailed textures, which is consistent with the existing literature.