[Reviews · NeurIPS 2020]

Review 1

Summary and Contributions: Normalizing flows are conventionally requires very large neural network models to obtain state-of-the-art results on density estimation benchmarks. This work proposes to reduce the amount of memory required by sharing most parameters among the internal flows. The authors apply their technique to the architecture of WaveFlow and a scaled-up version of Glow on audio generation and CIFAR10 image generation respectively. They find that naively sharing all parameters between internal flows reduces model's performance. Instead, they improve upon the naive method by adding a projection layer after each flow and adding layer dependent embeddings that modify the input, which they term NanoFlow. Their experiments show that NanoFlow performs slightly worse than the same architecture without parameter sharing but significantly reduces the number of parameters needed. ***************************************** Post rebuttal update: I appreciate the additional comparisons that the authors included in their rebuttal. My original concern was that there should be more extensive empirical evaluations in light of the incremental methodology. The new comparisons strengthen the empirical evaluations of the paper, and the negative results from the rebuttal's Table 2 helps the reader better understand the caveats of NanoFlow. The preliminary results and the promised results convinced me to raise my score to 6.

Strengths: 1.The paper addresses the question of how to maintain the same expressivity while reducing the memory required. This is a very relevant problem for normalizing flows since state-of-the-art-models typically require significant computational resources and time due to their size. 2. The method is simple and easy to understand. Prior work not related to normalizing flows has used similar parameter-sharing schemes to achieve good results. 3. The authors evaluate their method on two types of density estimation tasks, one for audio generation and one for image generation.

Weaknesses: Although the authors claim to have done "extensive" albation studies, the actual studies left much to be desired in terms of methodology and extensiveness. Since the method is simple, the paper should be judged based on how thoughtful and well-executed the experimental results were. The main weaknesses of the experiments are: 1. Insufficient evaluation of the baseline. As the authors noted, the "NanoFlow-decomp" is the most natural baseline beyond simply sharing every parameter between internal flows. However, the performance of "NanoFlow-decomp" only appears once in Table 4 when comparing to Glow. Given that "NanoFlow" is only a small modification of "NanoFlow-decomp", "NanoFlow-decomp" deserves much more attention to allow us to understand how much is gained by adding embeddings on top of "NanoFlow-decomp". It could be the case, for example, that "NanoFlow-decomp" would do almost as well as "NanoFlow", as suggested by their very similar bpd in Table 4. 2. Insufficient evaluation of design choices. The authors do not compare the effectiveness of different choices in their design for NanoFlow. For example, choices for the projection layer could be either fully connected layers or convolutions. Additionally, the authors do not empirically compare the different embedding choices outlined in line 135-137. 3. The amount of expressiveness lost by sharing parameters could be better evaluated. For example, given that we are sharing layers, how does performance scale with the number of layers shared? Currently, we only see K=8 and K=16 in Table 2 and K=32 and K=48 for Glow. 4. Potentially moving the goal post when making comparisons. Table 4 presents a questionable comparison between Glow-large and NanoFlow where they modify the baseline to compare their model. The authors claim that the architecture of Glow is not well suited for adapting to NanoFlow, so they implement a larger version of Glow, Glow-large, against the NanoFlow version of Glow. However, Table 4 never shows how the NanoFlow version of Glow would perform. Additionally, although Table 4 seems to suggest a 28x reduction in parameter for similar performance as Glow-large, the performance of the Glow architecture (which originally had 15M parameters) could have saturated well before reaching ~280M parameters. 5. Finally, although the paper aims to address model size generically, it only evaluates their method to one class of normalizing flows. A more extensive evaluation of their parameter sharing scheme would compare the performance on other flow architectures, not just flows based on affine coupling layers.

Correctness: Yes the claims are correct, but the experiments has methodology issues, as noted in the "weaknesses" section.

Clarity: The writing is generally easy to understand with some minor typos. However, the description of the methodology is sometimes vague which makes it hard to understand precisely what is done implementationally. For example: 1. They don't describe exactly what kind of projection layers were used in the experiments. Line 115 suggests 1x1 convolutions but it isn't explicitly stated 2. The authors also describe three candidates for embeddings in line 135-137 but the precise implementation for candidate 2 and candidate 3 is not very clear. Clarity in the "Broader Impact" section can also be improved.

Relation to Prior Work: Yes prior work is clearly discussed.

Reproducibility: Yes

Additional Feedback: Line 59 "systemic" -> "systematic" Line 136: "inside \hat \theta" Should this be "inside g^k"? Line 138: "inside \hat \theta" Should this be "inside g^k"? Line 152: "which evaluates under a condition where..." -> "which evaluates the conditions under which" Line 168: "WaveNet-based architecture" -> "the Wavenet-based architecture" Line 310: "as domain agnostic and independent variables," -> "is domain agnostic," Line 316: "the biases" -> "biases" Given how similar the LL/BPD are in Table 3 and 4, running multiple trials with standard deviations would help clarify sensitivity to initialization.


Review 2

Summary and Contributions: The paper proposes a parameter sharing scheme that can reduce the number of parameters of a normalizing flow generative model, while maintaining competitive performance. Concretely, the neural networks that predict the scale and shift of affine coupling/autoregressive layers are partially shared between bijectors. The parts that are not shared are the final layer of each neural network, as well as a learned embedding per bijector that serves as input to the shared bijector neural network. The idea is evaluated on audio and image data. For the audio domain the authors show that the parameter sharing method can create a model that outperforms a waveflow model with double the amount of parameters. However, the audio synthesis speed of the larger waveflow method is also roughly twice as large. In the image domain the proposed weight sharing strategy is tested on Cifar10 and compared with Glow. ---------------- Post-rebuttal update ------------------ I appreciate that the authors have taken significant effort to address the comments of the reviewers. My concerns about a more fair comparison against Glow and a comparison against neural spline flows and continuous flows have been addressed in the rebuttal. I have therefore increased my score to a 6.

Strengths: 1. The paper presents a clear research goal: reducing the parameter count of normalizing flow models without compromising too much on performance. This is an important research direction as it is required to make normalizing flows more practical for density estimation or data generation on memory constrained devices. 2. The presented solution is simple and the results indicate that the method shows some improvements in parameter reduction.

Weaknesses: 1. The number of parameters in the author’s reimplementation of Glow that achieves a result close to the one reported in the original paper is much larger (+/-287M parameters for 3.34 bpd) than what is reported in the literature for a similar performance (+/- 44M parameters for 3.35 bpd, see Table 3 in [3]). This significant difference gives a potentially misleading impression of how much the parameter count really is reduced by the proposed sharing scheme. 2. There is significant room for improvement in terms of comparing the proposed method to related work. For instance, the parameter counts and bpd numbers of other methods such as neural spline flows [3], which also significantly reduces the parameter count, should be included in table 4. The argument that other approaches to parameter reduction are orthogonal to the approach of the authors does not mean the results cannot be compared. Similarly, continuous normalizing flows [1, 2] also use a form of parameter sharing, and have a significantly smaller memory footprint as compared to non-continuous normalizing flows. This method is not mentioned as related work, it is not compared to in table 4, nor is there any investigation into adapting some of the weight sharing strategies used by these methods. 3. The paper writing can be improved, see more detailed comments about this in the “clarity” section. [1] https://arxiv.org/abs/1806.07366 [2] https://arxiv.org/abs/1810.01367 [3] https://arxiv.org/abs/1906.04032

Correctness: As mentioned above, the number of parameters of the baseline glow model (glow-large) is surprisingly large as compared to what is reported in the literature.

Clarity: The clarity of writing can be improved. The biggest improvement can come from being more consistent in notation and from referring more clearly to parts of the model/architecture. For instance, when referring to the shared network g with parameters theta in section 3.2, the authors refer to “each layer inside theta”, where I assume the authors mean “each layer inside g”. Upper and lower case symbols are inconsistently used to denote random variables and their values. Furthermore, section 3.1 refers to f^k_{mu, theta}, which I think should be f^k_{mu, sigma} according to the notation in Table 1.

Relation to Prior Work: As mentioned above, the comparison with related work has room for improvement.

Reproducibility: Yes

Additional Feedback: 1. The background section on normalizing flows does not explain that there are many other flows that do not consist of affine transformations, such as [5]. 2. The authors mention that waveflow provided a unifying view of autoregressive flows and bipartite coupling layers, but it would be worth mentioning that [4], which was published before waveflow, has also done this. Please refer to both works at least when mentioning this. [4] https://arxiv.org/abs/1705.07057 [5] https://arxiv.org/abs/1902.00275


Review 3

Summary and Contributions: This work investigates the memory efficiency in flow-based models, which is important for scaling up these models for real-world applications but was mostly neglected in current normalizing flow research. It studies several parameter sharing mechanisms, including the novel NanoFlow approach. It is also quite interesting to investigate the proposed parameter sharing mechanisms within WaveFlow framework, which allows varying amount of autoregressive structural bias for flow-based models.

Strengths: - It investigates important problem. - The proposed method is novel. - The likelihood gain is noticeable with comparable amount of parameters.

Weaknesses: - The generative results (audios / images) are not compelling.

Correctness: Yes.

Clarity: Yes.

Relation to Prior Work: Yes.

Reproducibility: Yes

Additional Feedback: - The attached speech samples are worse than the original WaveFlow, which have been open sourced recently. The authors may check the implementation detail and improve the baseline accordingly. Also, the WaveFlow paper suggests to train smaller models with more iterations for better speech fidelity (e.g., 2M iterations), which totally makes sense because smaller models underfit badly. I think the suboptimal performance in this work may come from less iterations (1.2 M iterations). I recommend the authors to train smaller models (both WaveFlow-64 and NanoFlow) with more iterations for higher speech fidelity, which will strengthen the paper. - This work suggests that the model with the least autoregressiveness (e.g., RealNVP, Glow), will benefit most from the proposed parameter sharing method. This is particularly useful, as the Glow-like models require huge amount of parameters to reach comparable model capacity as autoregressive models. One may also mention that IAF-based models (e.g., parallel wavenet, clarinet) can do parallel synthesis and still have the same model capacity as autoregressive models, but the likelihood-based training would be impractical.


Review 4

Summary and Contributions: This paper proposes a method to reduce the size of flow based models. Most of the parameters of a density estimator are shared across flows. Therefore the model size is substantially reduced. The introduction of embedding and projection layer improves the model capacity with negligible model size increase.

Strengths: 1. The model size is clearly reduced by weight sharing. 2. Evaluation is done on image generation and speech synthesis tasks. The proposed model outperforms baseline in terms of quality while staying compact. 3. Ablation study is done for embedding and projection layer. 4. The proposed model has been evaluated on image density estimation and audio synthesis tasks. NanoFlow is shown to significantly outperform NanoFlow naive.

Weaknesses: 1. Although weight sharing across different flows reduces the model size, it does not improve inference speed. The proposed model still computes same number of flows. 2. Weight sharing with embedding and projection layer is not novel. The authors applies the method on flow based models. The paper is an incremental work.

Correctness: The claims are mostly correct. For the speech synthesis task, the authors tested the proposed model when the number of groups varies. A smaller number of groups leads to more autoregressive structure, and improves the expressivity of the proposed model. This evaluation is thorough.

Clarity: Yes, the paper structure looks good to me. The experiments are clearly documented. In particular, the authors provides in-depth analysis for audio synthesis evaluation. Autoregressive structure bias is studied.

Relation to Prior Work: Previous works have been sufficiently discussed. The proposed method uses weight sharing between different flows to dramatically reduce model size. To increase model capacity, the authors introduced flow-wise embedding layer and projection layers, both are small in terms of parameters.

Reproducibility: Yes

Additional Feedback: 1. It would be good to provide audio samples generated from different flows (first to last) to reveal the effect of each flow. 2. Please elaborate on the concept of bits per dim in image density estimation task.

[Author Response · NeurIPS 2020]

Thank you for the constructive feedback. We address concerns raised by the
reviewers 1 and 2 regarding comparative results by an update with additional
baselines in the literature.

**Updated results with baselines** Table 1 shows an updated version of the Glow-
based experiments. Our initial demonstration at submission timeline was an
ablation study without full convergence. This could potentially result in mislead-
ing impressions of our method. After the submission, we trained each models
for longer duration up to 3000 epochs where each model reached the saturation.
Thus, we can now accurately compare the models to the results in the literature.
† indicates that the numbers are taken from existing literature under uniform
dequantization regime for fairness (Finlay et al, How to train your neural ODE,
ICML 2020). Reviewer 1 and 2 raised a valid concern regarding to the baseline
Glow model that it is hard to identify the gain in bpd at the cost of the larger
architecture. Here, we accurately show that the baseline Glow does scale with the
higher capacity, at the cost of the increased parameters and diminishing return.
NanoFlow outperformed the recently proposed models with less capacity, even
compared to models with more complex non-affine coupling, including neural
spline flows (RQ-NSF (C)), Flow++, and ResidualFlow.

**Filling in missing results** Reviewer 1 pointed out that the results should also
include NanoFlow's method applied to the reference topology of Glow to fully
compare the results for better understanding of the method. Table 2 shows the
results when applied to the reference Glow topology. Original Glow uses a total
of 3 layers: $3 \times 3$ conv $\rightarrow 1 \times 1$ conv $\rightarrow 3 \times 3$ projection conv. NanoFlowAlt
shares the first two layers and use separate $3 \times 3$ projection conv. Results show
that the model performed significantly worse than the baseline architecture, even
though NanoFlowAlt (K=48) has similar network size (10 M) to our main result.
This is reasonable and expected, because the shared estimator's capacity is too
restricted (0.7 M) to model multistep densities. Thus, one should be careful
about allocating the parameters under NanoFlow framework, where the shared
estimator should have sufficient capacity, while keeping the non-shared projection
layers slim. We hope that these addional results combined with our main updated
result would alleviate the concern about "moving the goal post". Reviewer 1 also
mentioned that NanoFlow-decomp is missing in WaveFlow results, thus it is hard to identify how much the gain is
achieved from our two strategies (decomposition and embedding). Table 3 shows that NanoFlow-decomp stays between
NanoFlow-naive and NanoFlow, consistent with the Glow-based models. Thus, we emphasize that both methods do
contribute to the performance. We will add the converged results. We also attach preliminary results applying RQ-NSF
to WaveFlow-based models in Table 3, and show that NanoFlow is applicable beyond affine coupling.

**Implementation details and choice of embedding** We will precisely add im-
plementation details of the embedding strategies we used to improve clarity. The
input at the start of each flow is $h^{k,0} = concatenate(x, \epsilon^k)$. for $l$-th layer inside
$k$-th flow, we get hidden state for the next layer $h^{k,l+1}$ as follows:

$$h^{k,l+1} = \exp(\delta^{k,l}) * (g^{k,l}(h^{k,l}; \hat{\theta}^{\cdot,l}) + g^{k,l}_{embed}(\epsilon^k; \eta^{k,l})) \tag{1}$$

where $\delta^{k,l} \in \mathbb{R}^H$ serves as the multiplicative gating and $H$ is the num-
ber of hidden channels. It is initialized to zero to initially perform identity.
$g^{k,l}_{embed}(\epsilon^k; \eta^{k,l}) \in \mathbb{R}^H$ is the additive bias obtained by $\eta^{k,l}$ with one fully-
connected layer ($1 \times 1$ convolution). $\eta^{k,l}$ is discarded after training and caching
the bias vectors from $g^{k,l}_{embed}$. We observed that these implementation choices
ensured stability during training. We partially dropped the methods that showed
no improvements during preliminary experiments, depending on the architecture.

**Improving related work** In line with the updated results in Table 1, we will
describe other classes of flows beyond the affine coupling in related work and background. Notably, continuous-time
flows (CNF) can utilize a "shared" neural network $f$, as commented by reviewer 2. The central difference between CNF
and NanoFlow (and non-continuous NFs in general) is that CNFs use numerical ODE solvers that iteratively evaluate $f$
to reach below tolerance, whereas NFs directly model pre-defined steps of transformation with $f_k$ (or $f$ in NanoFlow)
with a single pass. The effectiveness and potential benefits of the shared $f$ outside the ODE-based CNFs are yet to be
studied in the literature, where NanoFlow aims to systematically address, reaching to non-trivial solutions as proposed.

Table 1: Update of Table 4 with con-
verged models and additional base-
lines.

| Model (converged) | Params (M) | bpd |
|---|---|---|
| Glow (256 channels) | 15.973 | 3.40 |
| Glow (512 channels) † | 44.235 | 3.35 |
| Glow-large | 287.489 | 3.30 |
| RQ-NSF (C) † | 11.8 | 3.38 |
| FFJORD † | 0.801 | 3.40 |
| Flow++ † | 32.3 | 3.28 |
| ResidualFlow † | 25.174 | 3.28 |
| NanoFlow-naive | 9.263 | 3.40 |
| NanoFlow-decomp | 9.935 | 3.32 |
| NanoFlow | 10.113 | **3.27** |
| NanoFlow (K=48) | 10.718 | **3.25** |

Table 2: Results when applying the
method to the reference topology of
Glow model (NanoFlowAlt), evalu-
ated at 600 epochs.

| Model (600 epochs) | Params (M) | bpd |
|---|---|---|
| Glow (256 channels) | 15.973 | **3.44** |
| NanoFlowAlt-naive | 0.778 | 3.75 |
| NanoFlowAlt-decomp | 6.783 | 3.54 |
| NanoFlowAlt | 6.961 | 3.53 |
| NanoFlowAlt (K=48) | 10.319 | 3.51 |

Table 3: Additional WaveFlow
experiment, now with NanoFlow-
decomp evaluated at 100K steps.

| Model | Params (M) | LL |
|---|---|---|
| WaveFlow | 22.336 | 5.1164 |
| NanoFlow-naive | 2.792 | 5.0341 |
| NanoFlow-decomp | 2.794 | 5.048 |
| NanoFlow | 2.818 | 5.0774 |
| WaveFlow + RQ-NSF | 22.432 | 5.1262 |
| NanoFlow + RQ-NSF | 2.915 | 5.0862 |

[Meta-Review · NeurIPS 2020]

The paper investigates ways to reduce the parameter footprint of high-performant normalizing flows in image and audio modelling. The paper points out that naively tying parameters across layers doesn't work well, and so it proposes a better-thought way of doing this. The paper tackles an important problem - reducing the parameter footprint of large flow models can have significant benefits both in production (reducing the computational and memory footprint of these models) and in research (making these models easier to train and experiment with). Since the paper's contribution is an engineering improvement motivated by intuitive arguments rather than a rigorous theoretical justification, the reviewers correctly point out that a rigorous experimental validation is needed. Originally the reviewers had concerns regarding the rigour and depth of the experimental evaluation, but the additional experiments presented in the rebuttal have convinced the reviewers that the experimental evaluation is sufficient. I would strongly encourage the authors to take to heart the reviewers' feedback when revising the paper, in particular regarding strengthening the experimental evaluation, and to include the additional experimental results in the revised version.